# Brief Communication: towards a universal formula for the probability of tornadoes

Roberto Ingrosso[1], Piero Lionello[2], Mario Marcello Miglietta[3], and Gianfausto Salvadori[4]

[1]Department of Earth and Atmospheric Sciences, University of Quebec in Montréal, 201, av. du President Kennedy, Montréal,H3C 3P8,Canada
[2]Dipartimento di Scienze Ambientali e Biologiche, Università del Salento, via per Monteroni 165, Lecce, 73100, Italy
[3]ISAC-CNR, Istituto di Scienze dell'Atmosfera e del Clima, Consiglio Nazionale delle Ricerche, corso Stati Uniti 4, Padua, 35127, Italy
[4]Dipartimento di Matematica e Fisica, Università del Salento, Provinciale Lecce-Arnesano, P.O.Box 193, Lecce, 73100, Italy

**Correspondence:** Piero Lionello (piero.lionello@unisalento.it)

**Abstract.** A methodological approach is proposed to provide an analytical (exponential-like) expression for the probability of occurrence of tornadoes as a function of the convective available potential energy and the wind shear (or, alternatively, the storm relative helicity). The resulting expression allows to compute the probability of tornado occurrence using variables that are computed by weather prediction and climate models, thus compensating for the lack of resolution needed to resolve these phenomena in numerical simulations.

## 1 Introduction

Tornadoes are rapidly rotating columns of air (American Meteorological Society, 2020), extending vertically from the surface to the base of a cumuliform cloud, and represent one of the most severe weather phenomena in terms of victims and damages. Considering only the USA, every year about 500 tornadoes (Kunkel et al., 2013) of intensity EF1 (Enhanced Fujita scale, Fujita (1971); Potter (2007)) or stronger occur, producing an average of 125 victims and huge devastation (Ashley, 2007). Numerical simulations of the very fine spatial and temporal scale of tornadoes (typically with a diameter of less than 2 km and a duration of less than 1000 s) require resolutions that are orders of magnitude smaller than those currently available in operational weather prediction and climate models (Yokota et al., 2018). Further, the chaotic dynamics of these vortices limit their deterministic prediction (Markowski, 2020). Consequently, climatological studies focused on the identification of the environmental conditions favourable to tornado-spawning severe convective storms. Several thermodynamics and kinematic meteorological parameters have been analysed, either individually or considering combined instability indices, to identify the conditions most favourable to the genesis of tornadoes (Brooks et al., 2003; Romero et al., 2007; Taszarek et al., 2018, 2020; Ingrosso et al., 2020; Bagaglini et al., 2021). This approach is consistent with the basic idea that tornadoes result from a multi-stage process, which takes into account that the tilting of the horizontal vorticity near the ground by a violent updraft plays a basic role (Rotunno, 2013; Davies-Jones, 2015). Such a conceptual model is used here as a background framework for introducing an analytical formula for the probability of tornado occurrence. A previous study defined a tornado index

limited to the USA based on a Poisson regression between the observed U.S. climatology of tornadoes and monthly averaged environmental parameters from reanalysis (Tippett et al., 2012). Other studies limited their conclusions to the identification of the conditions that are associated with mesoscale convective hazards (Brooks, 2013; Diffenbaugh et al., 2013). The expression that we propose in this study is meant to provide a tool for supporting tornado warning in operational weather predictions and estimating changes of frequency of tornado occurrence in climate projections.

## 2   Data and Methods

Our analysis is based on tornadoes that occurred in the USA (dataset provided by the Storm Prediction Center-SPC, https://www.spc.noaa.gov/wcm/#dat) and in Europe (dataset provided by the European Severe Weather Database (ESWD), https://www.essl.org, managed by the European Severe Storm Laboratory (ESSL), Dotzek et al. (2009)). We considered only tornadoes of category 2 or higher (F2+), following the idea that weak events might have an uncertain signature in the environmental conditions and their reporting in official databases is less accurate. A total number of 3073 tornadoes have been considered in this study (2632 for the USA and 441 for Europe, see Supplementary Material for density plots) during the period 2000-2018. Unfortunately, our dataset does not allow us to differentiate supercellular tornadoes from landspouts in most cases. The hourly fields of ERA5 (ECMWF ReAnalysis 5, (Hersbach et al., 2020)) are used to establish a statistical link between the occurrence of tornadoes and a set of meteorological variables, allowing a straightforward physical interpretation of the results: the updraft maximum parcel vertical velocity (WMAX), which depends on the Convective Available Potential Energy CAPE, the mid-level wind shear ($WS_{700}$), the low-level storm relative helicity ($SRH_{900}$), and the lifting condensation level (LCL, Kaltenböck et al. (2009)). The Supplementary Material reports the expressions defining the variables used in this study. The values of these variables have been extracted in the period 2000–2018 in all cells where at least one tornado occurred, considering the hourly reanalysis fields at 25 km resolution. The values corresponding to the occurrence of tornadoes have been selected considering the time step closest to the recorded time of the tornado onset in the database.

The univariate analysis of the (conditional) probability $P$ of tornado occurrence is carried out by partitioning the observed range spanned by each variable into 17 equi-probable sub-intervals (bins). Such a number has been chosen as a compromise between the need of a number of bins sufficient for robust regressions and of a number of observations in each bin sufficient for a robust statistical analysis. An empirical estimate of the probability of tornado occurrence, conditional to the fact that the value of the variable lies in a given bin, is computed as the relative frequency of tornadoes in the bin. Its uncertainty is estimated via a suitable Bootstrap (Monte Carlo) procedure. An analytical expression of $y = log_{10}P$ is found by a simple linear regression for $WS_{700}$, $SRH_{900}$, and LCL, and by a non-linear regression for WMAX (see the Supplementary Material). Notice that, first the climatology of the variable of interest is calculated via the partition mentioned above, and then it is compared with the tornadic cases (an approach similar to the one adopted in Romero et al. (2007)).

## 3  Results

The univariate analysis shows that all the four variables considered in our study (i.e. WMAX, $WS_{700}$, $SRH_{900}$, LCL) are significantly linked to the formation of tornadoes. However, the formulas involving $WS_{700}$ and WMAX, i.e.

$$\log_{10} P = -6.8 + 0.11\,WS_{700} \tag{1}$$

$$\log_{10} P = -6.9 + \frac{WMAX}{3 + 0.32\,WMAX} \tag{2}$$

describe a range of probabilities (from $10^{-7}$ to $10^{-4}$) wider than that spanned by $SRH_{900}$ and LCL. In the case of $WS_{700}$, the probability increases exponentially over the whole range. Instead, the behaviour of $\log_{10} P$ as a function of WMAX is non-linear and shows a hyper-exponential increase of $P$ for low values (WMAX < 10 m/s), when the probability is small (about $10^{-7}$); in the intermediate range the growth gradually slows down, and $P$ becomes quasi-constant for large values (WMAX > 30 m/s), where the probability tends to $\approx 10^{-4}$. For LCL and $SRH_{900}$, the exponential decrease and increase, respectively, only describe a narrow range of probability (approximately from $10^{-6}$ to $10^{-5}$). In other words, variations of these two variables do not allow to discriminate among low and high probability of occurrence of tornadoes as effectively as in the case of $WS_{700}$ and WMAX (see Fig. 1).

Concerning the bivariate analysis (i.e., considering the joint behavior of pairs of predictors), in analogy with the univariate case, a 17×17 grid matrix is constructed to partition the whole two-dimensional domain in cells. The empirical estimate of the (conditional) probability $P$ of tornado occurrence, provided that the pair of variables lie in a given cell, is empirically computed as above via the estimate of the relative frequency of occurrence. Six different bivariate analyses are carried out considering all possible pair combinations of WMAX, $WS_{700}$, $SRH_{900}$ and LCL. For the bivariate probability, non-linear expressions have been adopted for all the pairs of variables involving WMAX, and a multiple linear expression for the remaining pairs (see the Supplementary Material). The values of the parameters of the bivariate probability functions have been estimated by a regression of the proposed expressions over the empirical probabilities.

Considering the bivariate expression of $P$ as a function of the pairs (WMAX, LCL) and ($WS_{700}$, $SRH_{900}$), the second variable lacks significance, meaning that it provides information analogue to the first one of the pair (in fact, they are fairly correlated), but the first variable provides more (univariate) informative details than the second one in terms of the range of $P$. Considering the pairs (WMAX, $SRH_{900}$), ($WS_{700}$, LCL) and ($SRH_{900}$, LCL), the probability of tornadoes significantly depends on both variables, but they describe variation of $P$ only over 2–3 orders of magnitude, whereas using the pair (WMAX,$WS_{700}$) shown in Fig. 2 it is possible to discriminate between conditions where the probability ranges from $10^{-7}$ to $10^{-3}$ (see the Supplementary Materials section for the figures regarding all the other pairs). In conclusion, a valuable fit of the probability of occurrence of tornadoes over the range $10^{-7}$–$10^{-3}$ is

$$\log_{10} P = -6.6 + \frac{WMAX}{3.1 + 5.2\,WMAX/WS_{700}} \tag{3}$$

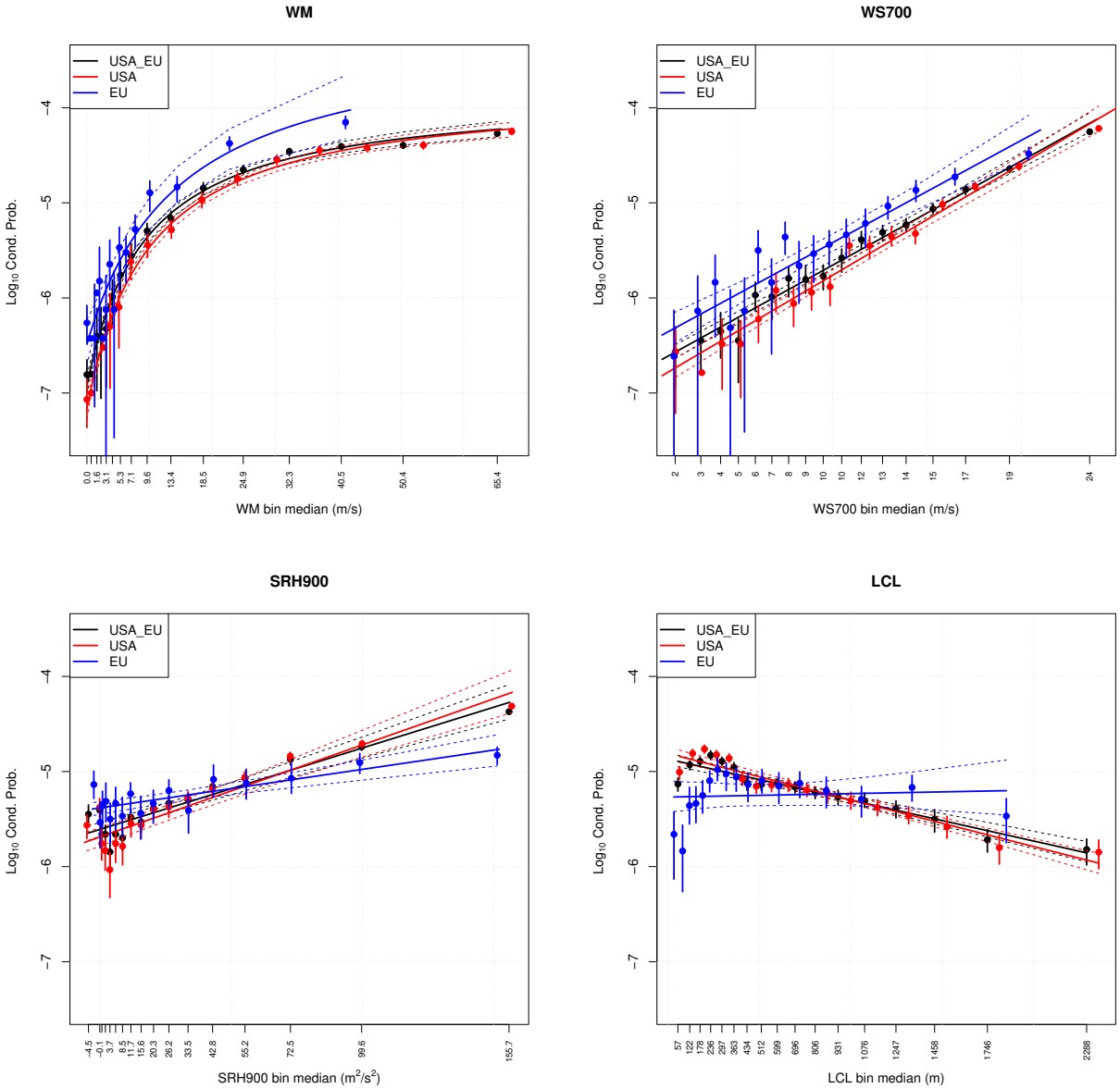

**Figure 1.** Univariate probability distribution for WMAX, WS$_{700}$, SRH$_{900}$ and LCL. Markers and whiskers denote the empirical probabilities with uncertainty range. Lines denote the empirical estimates (continuous) with uncertainty ranges (dashed). Different colours represent values based on the full dataset (USA&EU, black), the USA data only (red), and the European data only (EU, blue). Uncertainty ranges correspond to a 95% confidence level.

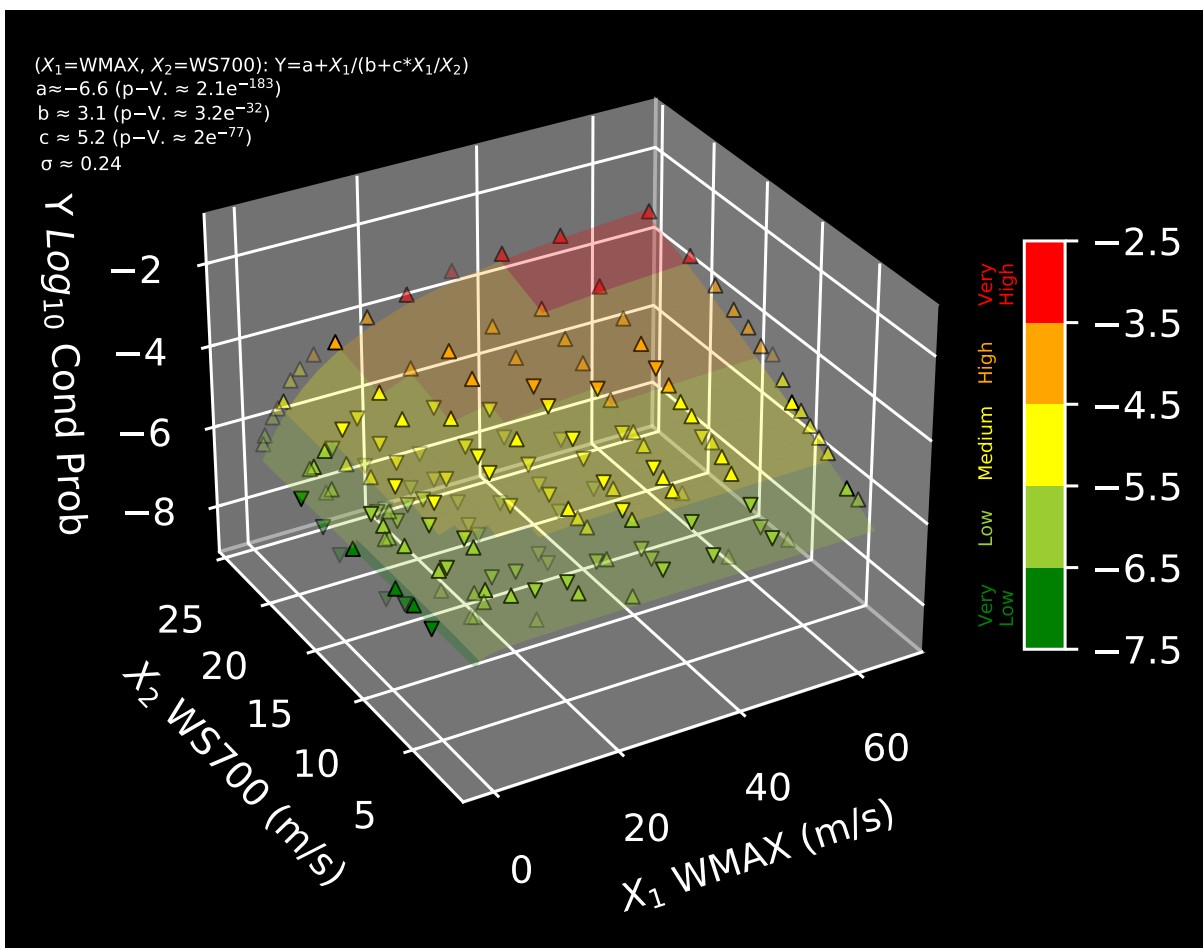

**Figure 2.** Bivariate probability distribution for $(X_1 = WMAX, X_2 = WS_{700})$. The coloured surface shows the empirical fit of $y = log_{10}P$. Upward/downward triangles represent empirical estimates located above/below the fitted surface. All values are reported according to the colour bar.

All parameters of the univariate fits in Fig. 1 and bivariate ones in Fig. 2 are statistically significant and significantly different from zero, since the p-values of the corresponding tests are (much) smaller than 1%. For all univariate linear regressions, the adjusted $R^2$ is larger than 90%, and, in general, the goodness of the fits is visually confirmed by the overwhelming fraction (from 90% to 100%) of probability values within the 95% confidence bands. In the bivariate case, considering the multiple linear regressions of the pairs (WS$_{700}$, SRH$_{900}$), (WS$_{700}$, LCL), and (SRH$_{900}$, LCL), $R^2$ is, respectively, 70%, 72%, and 54%: in general, these are smaller than in the single-variable case, but this is justified by the fact that the residual variances are about three times larger than those estimated in the univariate case. For the three pairs involving WMAX, $R^2$ cannot be used to assess the goodness-of-fit because the regression is non-linear. However, a slice-analysis of the fits (see the Supplementary Material for details) shows that the proposed models provide valuable fits over the whole domain of interest.

## 4   Discussion

Further investigations are required to ensure the validity of the expressions in Eqs. 1, 2, and 3 in different environmental and geomorphological conditions. Hypothesis testing the similarity of the populations of tornado probabilities $P_{\text{EU}}$ and $P_{\text{USA}}$, obtained using only EU and only USA data, respectively, has been carried out by using a Kolmogorov-Smirnov-like (KS) approach (Lopes, 2011) adopting the metric $d_0 = max|P_{EU} - P_{USA}|$. The significance level of the difference is assessed by computing the fraction of statistics exceeding $d_0$ using a Monte Carlo permutation procedure. Considering the univariate models, the null hypothesis that $P_{\text{EU}}$ and $P_{\text{USA}}$, as a function of WMAX and WS$_{700}$, are statistically compatible cannot be rejected at 95% and 99% levels (suggesting that Eqs. 1 and 2 are acceptable in different geographical domains), whereas it is rejected at a level larger than 99% for $P_{\text{EU}}$ and $P_{\text{USA}}$ as a function of SRH$_{900}$ and LCL. Considering the bivariate conditional probabilities, the null hypothesis - that $P_{\text{EU}}$ and $P_{\text{USA}}$ are statistically compatible - could not be rejected (at a 90% level) only for the pair (WMAX, SRH$_{900}$). In this case, the overall conditional probability (combining USA and EU data) is:

$$\log_{10} P = -6.6 + 0.34 \, WMAX^{0.37} |SRH_{900}|^{0.12} \tag{4}$$

For all other pairs the null hypothesis could be rejected at the 99% level.

Possible explanations of the lack of compatibility between conditional probabilities obtained using the EU and USA datasets alone could be: different tornadoes damage reporting practices (leading to different counting and attributions of tornadoes to the EF/F scale), and different meteorological and/or morphological conditions in the two domains. In spite of these limitations, and the need for further investigations, the proposed statistical models suitably fit the conditional probabilities of tornado occurrence. In particular, Eq. 3 has the merit of fitting the bulk of all available data, and Eqs. 1, 2 and 4 of being robust with respect to the considered geographical domains.

The formulas of Eqs. 1-4, and particularly the bivariate expressions of Eq. 3 and Eq. 4, outline a new statistical tool that can be used for diagnosing the likelihood of tornadoes with potential applications to short-medium range weather predictions and future changes of their frequency in climate projections. Former results considered monthly average probability (Tippett et al., 2012), or provided a modest fit to the data and were based on a smaller dataset (Cohen et al., 2018). The closest analogue to our approach is the formula of tornado probability in Grieser and Haines (2020), who considered two parameters: one describing vertical changes of temperature and a composite parameter merging CAPE and wind shear. Our results differ from Grieser and Haines (2020) in the adopted methodology for estimating the probability of occurrence of tornadoes. Grieser and Haines (2020) propose a linear regression of the logistic function, whereas we propose a nonlinear bivariate fit of the logarithm of the probability. In addition, our study shows that the relationship of CAPE to the probability of tornado occurrence departs significantly from a linear dependence, and that the interaction between the action of CAPE and wind shear in the lower troposphere cannot be adequately represented by their additive combination, further expanding the outcomes of Grieser and Haines (2020). Finally, Grieser and Haines (2020) used their formula for estimating past occurrence rates of tornado occurrences, while, to our best knowledge, this is the first time that analytical expressions in the form of eqs. (3) and (4) are

proposed in the scientific literature with the general aim of describing probability of tornadoes at high time and space resolution
125   with application in weather forecasting and climate projections.

*Sample availability* The list of tornadoes in the USA can be freely downloaded at https://www.spc.noaa.gov/wcm/#dat The list of tornadoes in Europe have been obtained from https://www.essl.org. ERA-5 data can be freely downloaded from https://cds.climate.copernicus.eu/cdsapp#!/home

*Author contributions.* RI has been responsible for data collecting, processing and plotting, PL for the coordination of the study, MMM for
130   the meteorological analysis, GS for the statistical analysis and the computation of the probability of occurrence of tornadoes. All the Authors wrote and contributed to the final manuscript.

*Competing interests.* PL is Editor for this journal

*Acknowledgements.* The Authors gratefully acknowledge useful discussions and suggestions by Prof. F. Durante (University of Salento, Lecce, Italy. The work of Piero Lionello has been carried out with the partial financial support from ICSC – Centro Nazionale di Ricerca in
135   High Performance Computing, Big Data and Quantum Computing, funded by European Union – NextGenerationEU (CUP F83C22000740001). Moreover, we thank the support of the European COST Action CA17109 'DAMOCLES" (Understanding and Modeling Compound Climate and Weather Events) and the support of the Italian PRIN 2017 (Research Projects of National Interest) "Stochastic Models of Complex Systems" [2017JFFHSH] are acknowledged. ESSL is acknowledged for providing European data; ECMWF for ERA5 reanalyses, Storm Prediction Center for US reports.

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
