# Peer review of "Brief Communication: towards a universal formula for the probability of tornadoes"

_Natural Hazards and Earth System Sciences, 2023_

## Author Response (AR1)

We thank the Reviewers for the thorough reviews and constructive comments that have helped us to improve the manuscript. Below, the reviewers' comments are in bold followed by the description of how each of their concerns has been addressed in the revised manuscript. The main changes made to the revised manuscript are highlighted in red.

Reviewer 1

**Comment: First of all, the text does not mention the difference between tornadoes associated with mesocyclones (.i.e. supercells) and those that are not (the so-called non-supercell tornadoes [1], or more precisely non-mesocyclonic tornadoes). Parameters used in the study to characterize the environmental conditions may differ substantially in both cases, and the choice of discarding weak (E/F0 and E/F1) tornadoes may not be useful to select only supercell tornadoes as non-supercell tornadoes may also be strong, as reported in the past [2]. This aspect should be at least mentioned. The text should also clarify if waterspouts are discarded in the datasets considered.**

Answer: We agree with the Reviewer that a distinction between mesocyclonic and non-mesocyclonic tornadoes would add an important piece of information to the paper. Unfortunately, the datasets used here provide this information only incompletely. On the other hand, the presence of E/F2+ tornadoes should guarantee the presence of a large majority of supercellular tornadoes in our analysis. In a forthcoming study, we are working to distinguish the environmental conditions favorable to mesocyclonic and non-mesocyclonic tornadoes in Italy, and in our dataset (2007-2022) all 36 E/F2+ tornadoes are identified as mesocyclonic.

We mention this point in the new version of the paper at the end of Line 34: "Unfortunately, our dataset does not allow us to differentiate supercellular tornadoes from landspouts in many cases.

**Comment: Secondly, did the authors consider the possibility of using records of non-tornadic events to assess the potential false alarms associated with the use of the results obtained? For example [3] calculated first climatologies of the parameters and then compared them with tornadic cases or [4] considered a sample of dry days and non-tornadic thunderstorm days to see if thresholds of the selected parameters associated with tornado occurrence differed substantially. I'm not asking that authors recalculate their computations but at least mention or discuss briefly these alternative approaches.**

Answer: Thanks for the suggestion, we decided to add this sentence to line 49-51 (end of Data and Method section).

*"Notice that, first the climatology of the variable of interest is calculated via the partition mentioned above, and then it is compared with the tornadic cases (an approach similar to the one adopted in Romero et al., 2007)."*

Romero, R., Gayà, M., & Doswell III, C. A. (2007). European climatology of severe convective storm environmental parameters: A test for significant tornado events. Atmospheric Research, 83(2-4), 389-404.

**Comment: Finally, the manuscript is submitted as a "Brief Communication", which according to the NHESS guidelines for authors means that it should have 2 to 4 pages, shorter than research articles (6 to 24 pages). Moreover, Brief Communications should have maximum 20 references, a number already exceeded. On the other hand, a description in the body of the text of the manuscript of the formulas of the parameters used and more details of the methodology are missing, which authors present in the "Supplementary Materials". I sincerely feel that readers would appreciate to find that information in the text. Thus, I strongly recommend that authors consider the possibility of expanding the current body of the text by moving there part of the "Supplementary Materials" and transform the "Brief Communication" into a research article.**

Answer: We have adopted the "brief communication" option because it is a more effective and faster tool for disseminating focused information on key novel theoretical results. In our case we perceived the urgency of providing a reference to the scientific community about results that have been already presented in international conferences (e.g. EGU22-11732 https://doi.org/10.5194/egusphere-egu22-11732) and can be readily used in research studies. The supplementary material of this manuscript will be available to readers and the transfer of corresponding material to the main body of the article would not actually change the available information. At this stage, abandoning the "brief communication" format would require a new submission and imply a significant delay for making the information immediately available to the scientific community. This, in our opinion, would be detrimental to the timely use of our results. The interest of the scientific community on extreme weather events, the associated risks and how they are impacted by climate change are nowadays hot research topics and, if the information provided by our study is considered relevant, it would be useful to make it available now, without delays. Anyway, we are working on a paper with an application of our theoretical results to be submitted within the end of this year.

Specific Comments

**Page 1, line 11. Suggest: The very fine spatial -> Numerical simulation of the very fine spatial [or similar]**

Done

**Page 2, line 31. Please specify the tornado intensity scale considered (i.e., F or EF?).**

Done

**Page 3, line 56. Please check: over all the whole range: do you mean over the whole range?**

Yes, done. Thanks.

**Page 4, Figure 1 (also in page 5, Figure 2, and in the Supplementary Materials). To avoid confusions, I suggest you change the label of the first panel (currently 'WM') to 'WMAX'.**

Done

**Also, in Figure 1 (and elsewhere). Authors identify USA and European tornados with the letters USA (three letters) and EU (two letters), respectively. This is a minor detail but, for the sake of consistency, you may consider using USA and EUR (both with three letters) or US and EU (both with two letters).**

Done

**Page 6, line 104. Typo: this limitations -> these limitations**

Done

**Page 8, Reference Dotzek et al: please check the use of capital letters where needed: european ... eswd -> European ... ESWD**

Done, also changed ESSL in the same reference and us to US in Tippett et al.

**Page 8, Reference Kunkel et al. Typo in: knowledge :. -> knowledge :**

 Done

**Page S1, section 1.1. Please indicate the units of the main variables used (WMAX, etc.)**

Done

**Page S1, Equation S5: the text indicates that T and Td are at 2m but the subindex indicates 10m (as in Equation S3). Please check.**

Formula corrected. Thanks.

Reviewer 2:

**Why the authors didn't use combination parameters such as energy helicity index (EHI) and Significant tornado parameter (STP)? These parameters would provide a more appropriate probability of tornado occurrences.**

Answer: We agree that combined parameters may provide more accurate univariate correlations; however, one of the motivations for the paper is to find a statistical relationship between tornado occurrence and parameters of immediate meteorological significance, in order to provide a straightforward physical interpretation of the results. Moreover, with respect to the use of combination parameters, our method has the advantage to quantify the role of the single predictors as well as to determine the best combination among different covariates with WS-WMAX as the best couple of predictors.

At Line 36: "suitable meteorological variables" has been changed into "meteorological variables allowing a straightforward physical interpretation of the results"

**Comment: What are the advantages of the method in the present study compared to the methods in the previous studies? More clearly explanation would be useful for readers.**

Answer: The innovative aspect consists in the proposed analytical expressions of the probability of tornado occurrence as functions of suitable meteorological variables based on the analysis of both USA and European tornadoes. Since these variables can be computed from the standard outputs of meteorological and climate models for each model grid point at high frequency (typically hourly or tri-hourly), the tornado probability can be computed at the corresponding high time and space resolution. The most effective among the proposed expressions is eq. (3), where probability is a function of the windshear in the low troposphere and of the convective available potential energy. Eq. (3) allows the immediate diagnosis of the tornado probabilities at a given time (hour and day), which could be used,

- in early warning systems to deliver a hazard level in a scale from very low to very high (see fig.2)
- to produce diagnostic maps of tornadoes probability
- to estimate the change of tornado frequency in climate projections

Note that also eq. (4) could be used, but it has a lower capability to distinguish between conditions leading to high and low probability of tornadoes.

We propose to add a paragraph at the end of the "Conclusion" section with this text.

*Former results considered monthly average probability (Tippett et al 2012), or provided a modest fit to the data and were based on a smaller dataset (Cohen et al., 2018). The closest analogue to our approach is the formula of tornado probability in Grieser and Haines (2020), who considered two parameters: one describing vertical changes of temperature and a composite parameter merging cape and wind shear. Our results differ from Grieser and Haines (2020) in the adopted methodology for estimating the probability of occurrence of tornadoes. Grieser and Haines (2020) propose a linear regression of the logistic function, while we propose a nonlinear bivariate fit of the logarithm of the probability. Further, our study shows that the relationship of CAPE to the probability of tornado occurrence departs significantly from a linear dependence, and that the interaction between the action of CAPE and the wind shear in the lower troposphere cannot be adequately represented by their additive combination, further expanding the outcomes of Grieser and Haines (2020). Finally, Grieser and Haines (2020) used their formula for estimating past occurrence rates of tornado occurrences, while, to our best knowledge, this is the first time that analytical expressions in the form of eqs. (3) and (4) are proposed in the scientific literature with the general aim of describing probability of tornadoes at high time and space resolution with application in weather forecasting and climate projections.*

Tippett, M. K., A. H. Sobel, andS. J. Camargo (2012), Association of U.S. tornado occurrence with monthly environmental parameters, Geophys. Res. Lett.,39, L02801, doi:10.1029/2011GL050368

Cohen, A. E., J. B. Cohen, R. L. Thompson, and B. T. Smith, 2018: Simulating Tornado Probability and Tornado Wind Speed Based on Statistical Models. Wea. Forecasting, 33, 1099–1108, https://doi.org/10.1175/WAF-D-17-0170.1.

Grieser, J.; Haines, P. Tornado Risk Climatology in Europe. *Atmosphere* **2020**, *11*, 768. https://doi.org/10.3390/atmos11070768

**Comment: Since a lot of information is included in Supplementary Material, readers would struggle to understand the study. I recommend more information is included in the body text as possible the authors can.**

Answer: We have adopted the "brief communication" option because it is a more effective and faster tool for disseminating focused information on key novel theoretical results. In our case we perceived the urgency of providing a reference to the scientific community about results that have been already presented in international conferences (e. g. EGU22-11732 https://doi.org/10.5194/egusphere-egu22-11732) and can be readily used in research studies. The supplementary material of this manuscript will be available to readers and transferring the corresponding material to the main body of the article would not actually change the available information. At this stage, abandoning the "brief communication" format would require a new submission and imply a significant delay for making the information immediately available to the scientific community. This, in our opinion, would be detrimental to the timely use of our results. The interest of the scientific community on extreme weather events, the associated risks and how they are impacted by

climate change are nowadays hot research topics and, if the information provided by our study is considered relevant, it would be useful to make it available now, without delays. Anyway, we are working on a paper with an application of our theoretical results to be submitted within the end of this year.